# Elicitation of Rank Correlations with Probabilities of Concordance: Method and Application to Building Management

**DOI:** 10.3390/e26050360

**Published:** 2024-04-25

**Authors:** Benjamin Ramousse, Miguel Angel Mendoza-Lugo, Guus Rongen, Oswaldo Morales-Nápoles

**Affiliations:** 1Department of Hydraulic Engineering, Delft University of Technology, 2628 CN Delft, The Netherlands; 2Linesight, 75014 Paris, France

**Keywords:** Bayesian networks, concordance probability, building maintenance, expert judgment, dependence calibration

## Abstract

Constructing Bayesian networks (BN) for practical applications presents significant challenges, especially in domains with limited empirical data available. In such situations, field experts are often consulted to estimate the model’s parameters, for instance, rank correlations in Gaussian copula-based Bayesian networks (GCBN). Because there is no consensus on a ‘best’ approach for eliciting these correlations, this paper proposes a framework that uses probabilities of concordance for assessing dependence, and the dependence calibration score to aggregate experts’ judgments. To demonstrate the relevance of our approach, the latter is implemented to populate a GCBN intended to estimate the condition of air handling units’ components—a key challenge in building asset management. While the elicitation of concordance probabilities was well received by the questionnaire respondents, the analysis of the results reveals notable disparities in the experts’ ability to quantify uncertainty. Moreover, the application of the dependence calibration aggregation method was hindered by the absence of relevant seed variables, thus failing to evaluate the participants’ field expertise. All in all, while the authors do not recommend to use the current model in practice, this study suggests that concordance probabilities should be further explored as an alternative approach for the elicitation of dependence.

## 1. Introduction

Significant attention, both in academia and in practice, has been directed towards the development of new techniques and expertise in the construction processes of buildings. However, the ageing of (Western European) building stock has progressively sparked interest in maintenance and future developments in the field [1]. As a result, standard practices have evolved from corrective maintenance, where works are performed after the occurrence of a failure in order to bring a component back into a state where it can perform its intended functions [2,3], to preventive maintenance, where interventions are performed following a specific schedule [3,4].

Preventive maintenance (PM) was originally used to pre-emptively reduce or eliminate the deterioration of building components [5,6]. Rapidly, though, scholars and practitioners came to realize that certain components were replaced despite being in good condition, thus incurring unnecessary costs. Consequently, condition-based maintenance (CBM) gained momentum. In contrast with predetermined PM, interventions in CBM are planned based on the condition of the assets under management, which is assessed during periodic inspections [5,7]. While the scope and methodologies of these inspections vary across regions and sectors, they commonly rely on the sensory assessment of individual building components’ condition.

Because of the poor accessibility and the complexity of mechanical, electrical, and plumbing (MEP) systems, these sensory inspections are rarely sufficient to reliably evaluate the condition of their components [8]. Asset managers are thus compelled to obtain condition data through estimates, commonly based on a limited number of parameters (e.g., age and theoretical lifespan; see, e.g., [9]), or through appraisals from third parties, resulting in a poor integration of these data in the overall maintenance strategies. This tendency led MEP to be the building trade where the highest number of defects are reported [10]. Clearly, then, implementing new methods to estimate these components’ condition is key to improve buildings’ occupants’ comfort [11,12,13] and minimize repair costs, which can be substantial [10,14].

To that end, the present study investigates the applicability of Bayesian networks (BNs) for the estimation of MEP systems’ condition. Bayesian networks, which are probabilistic graphical models used to study probabilistic influence between random variables, were selected because their graphical structure facilitates interactions with practitioners and they robustly handle missing data [15,16,17]. These characteristics are essential in the context of MEP systems given the scarcity of historical condition data. To better address this lack of data, so-called Gaussian copula-based Bayesian networks (GCBN) are adopted in this research. Their formulation, detailed in the next section, enables the involvement of field experts for the quantification of the model through structured expert judgments (SEJ), as demonstrated by past implementations of GCBNs (e.g., [18,19,20]).

Whereas the elicitation of univariate distributions has been investigated in academia with great depth, the assessment of dependence remains a topic yet to be consolidated in SEJ literature. Therefore, this paper focuses on the development of a method for the assessment of (conditional) rank correlations by field experts, while less attention is devoted to the elicitation of the one-dimensional marginal distributions. In contrast with existing research, which has delved into the use of statistical [19,21] and conditional fractile estimates [18,20,22] approaches, the relevance of a third type of probabilistic assessment is hereby studied: probabilities of concordance. Given the assumptions underlying GCBNs, unconditional rank correlations can be retrieved from concordance probabilities using a set of closed-form relations, which are defined in Section 2.2.1.

The next section presents theory on GCBNs and related statistical concepts (Section 2.1), and introduces the methodology implemented in this paper to retrieve rank correlations from expert judgments (Section 2.2). Then, the case study selected for the implementation of the aforementioned elicitation method is presented (Section 3). In Section 4, the results of the consultations are presented and analyzed, resulting in a quantified network for air handling units. Lastly, the research’s findings are discussed and conclusions are drawn with regards to the research objectives formulated above (Section 5).

## 2. Material and Methods

### 2.1. Gaussian Copula-Based Bayesian Networks

BNs are directed acyclic graphs (DAG) composed of nodes and arcs. In these networks, nodes represent random variables, while arcs represent the probabilistic dependencies between these variables [23]. The immediate predecessors of a node $X_{i}$ are called *parents* and noted $pa(X_{i})$; conversely, $X_{i}$ is called a *child* node of the elements of $pa(X_{i})$.

In contrast with discrete BNs, which use conditional probability tables to quantify influence, Gaussian copula-based Bayesian networks deal with dependence from another angle: dependence between variables is associated with (conditional) rank correlations, whose values depend on the non-unique ordering of each variable’s parents, and bivariate copulas—particularly the bivariate Gaussian copula. In previous literature, Gaussian copula-based Bayesian networks are sometimes referred to as “Non-Parametric” Bayesian networks (NPBN). However, the use of parametric one-dimensional marginal distributions motivated the authors to refer to them as “Gaussian copula-based Bayesian networks”.

Copulas were introduced in [24] with Sklar’s theorem, which states the following:

**Theorem 1.** 
*Given a joint cumulative distribution function (CDF) $F(x_{1},\ldots ,x_{n})$ for random variables $X_{1},\ldots ,X_{n}$ with marginal CDFs $F_{1}\left(x_{1}\right),\ldots ,F_{n}\left(x_{n}\right)$, F can be written as a function of its marginals:*

$$F\left(x_{1},\ldots ,x_{n}\right)=C_{\theta }\left(F_{1}\left(x_{1}\right),\ldots ,F_{n}\left(x_{n}\right)\right),$$

*where $C_{\theta }\left(u_{1},\ldots ,u_{n}\right)$ is a joint distribution function with uniform marginals. Moreover, if each $F_{i}$ is continuous, then $C_{\theta }$ is unique, and if each $F_{i}$ is discrete, then $C_{\theta }$ is unique on $Ran\left(F_{1}\right)\times \ldots \times Ran\left(F_{n}\right)$, where $Ran(F_{i})$ is the range of $F_{i}$. $C_{\theta }$ is called a copula with parameter(s) θ.*


Several measures of dependence in copulas exist, with Pearson’s product moment correlation (ρ) being the most widely used. GCBNs, however, employ Spearman’s rank correlation (*r*). For infinite continuous populations with zero probability for ties, *r* is proportional to the difference between the concordance and discordance probabilities. Consider two independent vectors of random variables $\left(X_{1},Y_{1}\right)$ and $\left(X_{2},Y_{2}\right)$, where $\left(X_{1},Y_{1}\right)$ has a joint distribution $F_{X,Y}$ with marginal distribution functions $F_{X}$ and $F_{Y}$, and where $X_{2}$ and $Y_{2}$ are independent with marginal distributions $F_{X}$ and $F_{Y}$. Then,
(1)$$r=3\left(P\left[\left(X_{1}-X_{2}\right)\left(Y_{1}-Y_{2}\right)>0\right]-P\left[\left(X_{1}-X_{2}\right)\left(Y_{1}-Y_{2}\right)<0\right]\right)$$

For a bivariate copula *C*, Equation (1) is equivalent to r=12∫C(u,v)dudv−3. Throughout this research, only populations with zero probability for ties are considered. A correction for dealing with populations with ties (see [25]) has not been integrated in the present study nor in the associated software implementation.

The conditional rank correlation of $X_{i},X_{j}|X_{k},\ldots ,X_{z}$ is the rank correlation of $\left({\bar{X}}_{i},{\bar{X}}_{j}\right)$ where $\left({\bar{X}}_{i},{\bar{X}}_{j}\right)$ have the distribution $X_{i},X_{j}|X_{k}=x_{k},\ldots ,X_{z}=x_{z}$. When unambiguous, the notations $\rho_{X,Y}$ (for ρ(X,Y)) and $r_{X,Y}$ (for r(X,Y)) are used in the remainder of this paper. Likewise, conditional rank correlations $r(X_{i},X_{j}|X_{k},\ldots ,X_{z})$ are noted $r_{X_{i},X_{j}|X_{k},\ldots ,X_{z}}$ when possible. Unlike product moment correlations, which assess linear dependence between two variables, rank correlations provide a more general measure of monotonic dependence, rendering it independent of the marginal distributions [18,26].

In GCBNs, each edge is associated to a (conditional) Gaussian copula parametrized by a (conditional) rank correlation; for each term *i* with parents $\left\{i_{1},\ldots ,i_{k}\right\}$, the rank correlation associated with the edge $i_{k-j}\to i$ is
(2)$$\left\{\begin{array}{cc} r(i,i_{k})\; & j=0,\\ r(i,i_{k-j}|i_{k},\ldots ,i_{k-j+1})\; & 1\leq j\leq k-1. \end{array}\right.$$
The assignment is vacuous if $pa(X_{i})=⌀$. Then, the GCBNs’ main result, demonstrated in [27] and extended in [26], states the following:

**Theorem 2.** 
*Given the following conditions, the joint distribution of the n variables of a network is uniquely determined:*
*1.* 
*A directed acyclic graph (DAG) with n nodes specifying conditional independence relationships in a BBN;*
*2.* 
*n variables $X_{1},\ldots ,X_{n}$, assigned to the nodes, with continuous invertible distribution functions;*
*3.* 
*The specification (2), i = 1, …, n, of conditional rank correlations on the arcs of the BBN;*
*4.* 
*A copula realizing all correlations [−1, 1] for which correlation 0 entails independence.*

*and the conditional rank correlations (2) are algebraically independent.*


The Gaussian copula offers several advantages that make its use attractive in Bayesian networks. The bivariate Gaussian copula is defined as
(3)$$C_{\rho }\left(u_{1},u_{2}\right)=\Phi_{\rho }\left(\Phi^{-1}\left(u_{1}\right),\Phi^{-1}\left(u_{2}\right)\right),$$
where $\Phi_{\rho }$ is the bivariate standard normal CDF with product moment correlation ρ and $\Phi^{-1}$ the inverse univariate standard normal CDF. The Gaussian copula allows for significantly faster conditionalization of the joint distributions due to one of its intrinsic properties: for multivariate Gaussian distributions, all conditional distributions are also Gaussian. Additionally, closed-form relations between different measures of dependence (such as Pearson’s correlation ρ, Spearman’s rank correlation *r*, and Kendall’s τ) are known for this copula family. Such relations are particularly pertinent when attempting to compute rank correlations from other statistical quantities, such as probabilities of concordance.

### 2.2. Dependence Assessment

Similarly to discrete Bayesian networks, the construction of a GCBN involves two distinct steps: the definition of a DAG as well as the quantification of its parameters. Therefore, it is essential to gather information on both the marginal distributions and the (conditional) rank correlations, even in contexts where data on the variables of interest are limited. As illustrated in [28], the elicitation of one-dimensional distributions from expert judgments has been discussed extensively in literature. Therefore, this section introduces methods for the expert-based elicitation of rank correlations, underlining this paper’s focus on the quantification of **dependence**.

#### 2.2.1. Concordance Probabilities

Assessing correlation between two variables has proved to be a challenging task. Direct elicitation methods can take various forms, but are commonly classified in three approaches: (i) statistical approaches, (ii) conditional fractile estimates and (iii) probabilities of concordance [18,29,30]. In the first, experts directly provide rank correlations estimates or related quantities such as ratios of rank correlations [19,21]. In the second, experts provide conditional probabilities of exceedance, answering questions such as *“Suppose that variable X was observed above its $q^{th}$ quantile, what is the probability that Y will also be observed above its $q^{th}$ quantile?”*. From the results, the assessor can compute the associated (conditional) rank correlations, as described in [18]. Despite the popularity of this approach for quantifying GCBNs [18,19,22], computing rank correlations from exceedance probabilities has limitations. For instance, this approach requires knowledge of the marginal distributions by experts and is suitable when working exclusively with continuous variables. Therefore, this research investigates the applicability of probabilities of concordance. A probability of concordance ($P_{c}$) is defined as follows: given a bivariate population (X,Y), two independent realizations ($x_{A},y_{A}$) and ($x_{B},y_{B}$) are considered. Then:(4)$$P_{c}=P\left(\left(x_{A}-x_{B}\right)\left(y_{A}-y_{B}\right)>0\right)=P\left(x_{A}<x_{B}|y_{A}<y_{B}\right)=P\left(x_{A}>x_{B}|y_{A}>y_{B}\right).$$

To the best of the authors’ knowledge, no study has relied on probabilities of concordance for the elicitation of rank correlations for (GC)BNs. While the use of probabilities of concordance may be inadequate for investigating correlation of rare events [29], it is highly relevant for problems that involve physically intelligible variables. For instance, take *X* as the variable representing the weight of Dutch males between 18 and 50 years old, and *Y* representing the height of the same population. $P_{c}\left(X,Y\right)$ is then obtained by answering the following question:

“Two individuals A and B are randomly selected among Dutch males between 18 and 50 years old. Given that B is taller than A $\left(y_{A}\leq y_{B}\right)$, what is the probability that B weighs more than A $\left(x_{A}\leq x_{B}\right)$ ?”

If a respondent believes that *X* and *Y* are completely positively (resp. negatively) correlated, then they should provide a value of $P_{c}=1$ (resp. $P_{c}=0$), while $P_{c}=0.5$ indicates independence between *X* and *Y*.

As outlined in Section 2.1, relations exist to retrieve rank correlations from $P_{c}$. First, note that Kendall’s τ is a re-scaled version of the probability of concordance [31,32,33]:(5)$$\tau =2P_{c}-1.$$
Given the Gaussian copula assumption, closed-form relations also exist between Kendall’s τ, Pearson’s ρ and Spearman’s *r* [26,34]: (6)$$\begin{array}{cc} \rho & =sin\left(\frac{\pi \tau }{2}\right), \end{array}$$(7)$$\begin{array}{cc} r & =\frac{6}{\pi }arcsin\left(\frac{\rho }{2}\right). \end{array}$$

Figure 1 illustrates the non-linear relationship between $P_{c}$ and *r*, along with the relationship between $P(F_{X_{1}}\geq q=0.5|F_{X_{2}}\geq q=0.5)$ and the rank correlation for the Gaussian, Clayton, and Gumbel copulas. The rotated Clayton and Gumbel copulas were used to capture negative dependence. One observes that under the Gaussian copula assumption, $P_{c}$ and $P(F_{X_{1}}\geq 0.5|F_{X_{2}}\geq 0.5)$ are equivalent. In contrast, slight differences can be appreciated between exceedance probabilities for values of *r* below 0.30 and above 0.70 between the Clayton copula and the others. Conducting similar experiments with different values of q reveals a greater variability in values of *r* based on the copula chosen, as illustrated in Figure A1 with q={0.25,0.75}. For instance, $P(F_{X_{1}}\geq 0.75|F_{X_{2}}\geq 0.75)$ may take any value in the interval [0,1] for all three copulas (see Figure A1a), while this is not the case for $P(F_{X_{1}}\geq 0.25|F_{X_{2}}\geq 0.25)$, where the conditional probability is constrained to the interval [0.66, 1] (see Figure A1b).

Clearly then, eliciting rank correlations in the form of conditional probabilities such as $P(F_{X_{1}}\geq q|F_{X_{2}}\geq q)$ for q≠0.5 and for copulas other than the Gaussian bears several limitations. Moreover, eliciting rank correlations in the form of concordance probabilities is arguably more intuitive than through exceedance probabilities. Building on the example of the Dutch male population, obtaining $P(F_{X_{1}}\geq 0.75|F_{X_{2}}\geq 0.75)$ requires experts to answer the question: “Suppose that individual A is taller than 75% of the Dutch males between 18 and 50 years old, what is the probability that he also weighs more than 75% of the same population ?”—which is all but intuitive. As a result, probabilities of concordance may represent an alternative in similarly practical situations.

After retrieving unconditional rank correlations with Equations (5)–(7), *conditional* rank correlations can be computed recursively using partial correlations and the ordering of each variable’s parents. Indeed, under the normal copula assumption, partial and conditional correlations are equal, the former being defined in Equation (8) [35]. If $X_{1},\ldots ,X_{n}$ are random variables, the partial correlation of $X_{1}$, $X_{2}$ given $X_{3},\ldots ,X_{n}$ is
(8)$$\rho_{12;3,\ldots ,n}=\frac{\rho_{12;4,\ldots ,n}-\rho_{13;4,\ldots ,n}\;\rho_{23;4,\ldots ,n}}{\sqrt{\left(\left(1-\rho_{13;4,\ldots ,n}^{2}\right)\left(1-\rho_{23;4,\ldots ,n}^{2}\right)\right)}}.$$

As stated in Theorem 2, the (conditional) rank correlations linked to the arcs of a GCBN are algebraically independent and guarantee the construction of a valid correlation matrix. Because the elicitation of rank correlations is carried out sequentially, the range in which an unconditional rank correlation—and thus a concordance probability—can take values is not necessarily [−1,1]. Let us consider the graph in Figure 2. The first rank correlation to be quantified by a particular expert would be $r_{1,2}$; the second one $r_{1,3}$; and finally, $r_{2,3|1}$. For instance, if $r_{1,2}=0.5$ and $r_{1,3}=0.7$, one can easily verify using Equation (8) that $r_{2,3}$
*must* be in [−0.27,0.97] for $r_{2,3|1}$ to remain within [−1,1].

To facilitate the computation of the conditional rank correlations, the software *Matlatzinca* (v.1.0.0) was used [36]. In addition to automating the required operations, *Matlatzinca* indicates for each edge the range of mathematically acceptable unconditional rank correlations, as discussed in the previous paragraph. An in-depth presentation of the software and its features is laid in the next section.

The protocol implemented to retrieve individual experts’ opinions can be summarized in a set of elementary steps as follows:The expert assesses the probability of concordance $P_{c}\in \left[0,1\right]$;$P_{c}$ is converted to an unconditional rank correlation using Equations (5)–(7);The correlation coefficient is logged into *Matlatzinca*. If the respondent’s answer is mathematically acceptable, move to the next question and go back to step 1;Else, the expert is given the mathematically valid range for $P_{c}$. Because this range is directly affected by their answers to the previous questions, the experts may review and modify previous answers accordingly.

#### 2.2.2. Software

As stated in the previous section, *Matlatzinca* (v.1.0.0) was used to retrieve the conditional rank correlations. The software was developed by researchers of the TU Delft, The Netherlands, and is strongly based on PyBANSHEE (v.1.0), a Python-based open-source implementation of the MATLAB toolbox BANSHEE (v.1.3) [37,38,39]. *Matlatzinca* is used to schematize and quantify a dependence model, specifically the GCBN, and is accessible on https://github.com/grongen/Matlatzinca (accessed on 26 April 2023). Noteworthily, two methods absent in PyBANSHEE were added: (i) the option to enter an unconditional correlation and get the associated rank correlation, and (ii) the computation of the range of mathematically *acceptable* or *valid* unconditional correlations.

The current version of *Matlatzinca*’s graphical user interface (GUI), shown in Figure 3, consists of three main panels:The **drawing** panel. This is where the DAG representing the dependence structure of the BN is drawn. Notice that, as discussed in Section 2.1, the arcs provide information regarding the ordering of parents in the DAG.The **input** panel. It contains, on the left-hand side, the labels of the Nodes displayed in the drawing panel, which can be edited by the user. On the right-hand side, it presents the Edges and related measures of dependence. For the quantification of the arcs, users have two input options: Spearman’s conditional rank correlations (Conditional rank corr.) as well as unconditional rank correlations (Non-conditional rank corr.). The last column indicates the range of acceptable unconditional rank correlations, briefly discussed at the end of the previous section, which depends on the structure of the DAG and other values of the correlations. This column is updated as users provide values of (un)conditional rank correlations.The **correlation matrix** panel. In addition to their numerical value, each correlation coefficient is displayed with a circle whose diameter is proportional to its absolute value, and a colormap indicating the position of the coefficient on the [−1, 1] scale.

For a comprehensive presentation of the functionalities of the *Matlatzinca* software, the reader is referred to [36] and the references therein.

Given the software’s current design, *Matlatzinca* is exclusively suitable for the quantification of expert-based networks. For data-based models, a wide range of software for the implementation of Bayesian networks (e.g., Netica, Hugin) exist. However, only a few are compatible with GCBNs; we thus opted for a combination of UniNet Academic (LightTwist Software, Brunswick, 3065 Australia) (software in closed-access; see https://www.tudelft.nl/en/eemcs/the-faculty/departments/applied-mathematics/applied-probability/research/research-themes/risk/software/uninet/ (accessed on 28 March 2023).) and PyBANSHEE. Whereas the former’s GUI is practical when interacting with external stakeholders, PyBANSHEE offers more flexibility when performing analyses, as highlighted by the recent applications of the software [40,41].

### 2.3. Dependence Calibration

After collecting the individual assessments, these must be aggregated in a unique correlation matrix. Two types of methods are found in the literature: behavioral and mathematical [42]. On the one hand, behavioral methods aim to reach consensus between the experts [42]. However, these approaches may result in a situation where agreement between experts is either impossible, or leads to compromises that reflect none of the experts’ opinions [42,43]. On the other hand, mathematical methods attempt to overcome behavioral biases by combining individual assessments through a mathematical process subject to empirical control. Although most mathematical aggregation approaches consist in weighing together the experts’ judgments, their complexity varies greatly: from arithmetic and geometric means, to methods which account for experts’ performance, such as the classical model (or Cooke’s method, after [44]) [28,43].

Because Cooke’s method was not designed for scoring dependence assessments, another performance-based method was investigated: the dependence calibration (or d-calibration [19]). The latter has already been applied to a handful of real-life problems [20,45,46]. For the Gaussian copula, this quantity measures the “distance” between two correlation matrices. In the context of experts’ judgments, let $R_{m}$ be the empirically observed correlation matrix and $R_{e}$ an expert’s estimation of that correlation matrix. The d-calibration score $dCal_{e}$ is then defined as
(9)$$dCal_{e}=1-d_{H}\left(R_{m},R_{e}\right)=1-\sqrt{1-\frac{|R_{m}{|}^{\frac{1}{4}}{\left|R_{e}\right|}^{\frac{1}{4}}}{|\frac{1}{2}R_{m}+\frac{1}{2}R_{e}{|}^{\frac{1}{2}}}},$$
where $d_{H}$ is the Hellinger distance. The d-calibration score hence takes values between 0 and 1 (for $R_{m}=R_{e}$). In the context of this study, $R_{m}$ is a correlation matrix used for *calibration* purposes and therefore contains information on the seed variables chosen by the authors.

The d-calibration score has the following properties: (a) an expert will receive the maximum score if and only if she/he captures the observed dependence structure exactly; (b) an expert may receive a low score if, for instance, a high correlation between a pair of variables was expressed by the expert while this was not reflected in the true dependence structure (or vice-versa); and (c) a necessary condition for an expert to be highly calibrated is to sufficiently approximate the dependence structure of interest element-wise [46].

It is worth noting that, similarly to Cooke’s method, the identification of relevant seed variables to evaluate experts’ calibration can be a significant challenge to the elicitation process. Because of the limited resources available in this study regarding time, empirical data, and experts’ availability, the quantification of a network that encompasses all MEP systems based solely on experts’ judgments is practically unrealistic. Therefore, the following paragraphs introduce the case study adopted in this paper, for which this elicitation method is applied and commented.

## 3. Case Study

A wide array of air handling units (AHUs) are available on the market, all designed with a shared purpose: maintaining acceptable indoor air quality. Except for single-family housing, central air handling units are commonly used and placed on a building’s roof. Figure 4 illustrates the process by which indoor air quality is preserved: outdoor air is filtered, conditioned by coils for heating or cooling, and distributed in the room(s) through ducts. Simultaneously, polluted indoor air is extracted and (partially) evacuated from the building.

This section describes the process of building a GCBN to estimate the condition of air handling units’ components. In particular, Section 3.1 presents the design of the network’s graph structure, while Section 3.2 focuses on the implementation of the elicitation method to the case study.

### 3.1. Graph Structure

The first step in the construction of the GCBN is the definition of its graph. In order to estimate the condition of air handling units’ components, a set of factors influencing their deterioration ought to be identified. An exhaustive literature review was conducted in [48] and resulted in the selection of the variables: (i) the *AHU age* (in years), (ii) the *maintenance interval* between two consecutive interventions (in years) and (iii) the *Design & Construction quality* of the installation, defined by the scale shown in Table 1.

Then, the main elements composing an AHU must be identified, which clearly stand out from Figure 4: (v) *plumbing supply*, (vi) *electrical supply*, (vi) exhaust *fans*, (vii) heating and cooling *coils*, and (vii) the *filters*. In their study on gradual fault prediction, Ref. [49] limited their effort to defects related to the supply fan and the cooling coil, obtaining satisfactory results. However, the present research also investigates the relation between components. Consequently, all the aforementioned elements are included and grouped in the following variables: coils, fans, and filters. The decision to group components is knowingly oversimplistic and reflects the exploratory dimension of the research, whose focus is on the elicitation of experts’ judgments rather than the creation of a complex and accurate model. For practical purposes, the condition of these components—and the associated variables—is defined in accordance with the 1–6 scale of NEN 2767, the Dutch standard for building condition assessment, where 1 represents an ‘excellent’ condition and 6 a ‘very bad’ one; see [9,50].

Finally, a set of assumptions was formulated to define dependencies (*parent* → *child*) between variables of the graph:Because of their comparatively short lifespan, the condition of the filters and the coils are exclusively affected by the maintenance interval, i.e., *Maintenance interval* → *Filters* and *Maintenance interval* → *Coils*.The condition of the plumbing supply system (boiler, chiller) affects the coils as these elements are functionally interdependent: the warm or chilled water (or other fluid) from the plumbing system supplies the coils, i.e., *Plumbing supply* → *Coils*. Likewise, the electrical supply system exclusively interacts with the fans, i.e., *Electrical supply*→ *Fans*.Since the filters are responsible for reducing the number of particles entering the AHU, their failure allows for the accumulation of particles on the coils and thus speeds up their deterioration by corrosion, i.e., *Filters* → *Coils*.The condition of the fans can be impacted by the filters in at least two ways. First, polluted filters oblige the fans to exert more power to maintain the same perceived airflow. Secondly, particles that enter the AHU partially flow through the ducts where they accumulate, thus leading to reduced airflow and additional stress on the fans. Clearly, then, these components are interdependent, i.e., *Filters* → *Fans*.The AHU’s age and the Design & Construction quality of the installation both directly affect the coils and fans, i.e., *AHU Age* → *Coils*, *AHU Age* → *Fans*, *Design & Construction quality* → *Coils*, and *Design & Construction quality* → *Fans*.

The resulting graph is shown in Figure 5.

### 3.2. Quantification: Experts’ Judgments

Section 2 introduced the framework for the elicitation of rank correlations in GCBNs. In this section, the application of these methods to the case study is presented, including the list of participants, the questionnaire, and the seed variables used for dependence calibration.

#### 3.2.1. Individual Assessments

Similarly to the case of Dutch males’ weight, height, and age discussed in Section 2.2, the study of air handling units (and MEP systems as a whole) is based on physical quantities. Let *X* be the condition of the fans and *Y* the age of the AHU as defined previously. To retrieve the probability of concordance $P_{c}\left(X,Y\right)$, one needs the answer to the following question:

“Two buildings A and B are randomly selected among all non-residential buildings in the Netherlands. Given that the air handling unit in building A is more recent than in building B $\left(x_{A}\leq x_{B}\right)$, what is the probability that the fans are in better condition in building A than in building B $\left(y_{A}\leq y_{B}\right)$ ?”

Because the graph in Figure 5 contains ten edges, the first and main section of the questionnaire included ten questions similar to the one formulated above.

Given the nature of the questions and the topic of the research, the experts contacted must be familiar with heating, ventilation, and air conditioning (HVAC) systems and their deterioration. Forming a diverse group, for instance, with regards to experience and private/public employment, is believed to result in more representative elicited quantities [51]. Therefore, practitioners and scholars from the TU Delft as well as industry participants were welcomed to participate, regardless of their level of experience. Moreover, participants were required to have basic comprehension and expression skills in English given that the questionnaire/interviews were conducted in that language, a criterion that proved constraining for some (potential) respondents. The panel of participants consulted for the assessment of probabilities included five experts, whose details are laid out in Appendix B. In the remainder of the paper, the experts are referred as ‘Expert A’, ‘Expert B’, and so forth to ensure the unbiased interpretation of the results.

#### 3.2.2. Aggregation

The formulation of relevant seed questions is a challenging task when empirical data is scarce or simply absent [52]. As outlined previously, condition assessment data for MEP systems is not widely available. Consequently, a seed variable familiar to the experts, although unrelated to the topic of this paper, was selected: precipitation (with over 100 rain days per year, rain is rooted in the Dutch culture. Source: https://www.statista.com/statistics/1012831/number-of-rain-days-in-the-netherlands/ (accessed on 20 June 2023)).

Empirical data of hourly precipitation (*Dutch: uur som van de neerslag*), measured at three weather stations between the 1 January 2023 and 18 June 2023, were retrieved from the online database of the Koninklijk Nederlands Meteorologisch Instituut (*Dutch Royal Institute of Meteorology—KNMI*). For reference, the location of the stations is illustrated in Figure 6 (left). Because of their geographical proximity, precipitations at these locations are likely to be correlated, an assumption supported by historical data. The empirical (rank) correlation matrix of variables ‘Gilze-Rijen’, ‘Rotterdam’ and ‘Eindhoven’ is retrieved and shown in Section 4. The second part of the questionnaire included the seed questions related to the graph in Figure 6 (right) and were formulated as follows:

“Two moments H1 and H2 (defined by the hour) are taken randomly between the 1 January 2023 and the 18 June 2023. Given that the hourly precipitation is higher at H2 than at H1 in Gilze-Rijen, what is the probability that the hourly precipitation is also higher at H2 than at H1 in Rotterdam?”

The questions presented to the experts were answered following the same protocol as the ‘main’ questions, presented in Section 2.2. The resulting correlation matrices were then used to compute individual experts’ d-calibration scores and are discussed in the next section.

#### 3.2.3. Marginal Distributions

Having dedicated extensive time and effort to the elicitation of the network’s dependence structure, a less scientifically sound approach was adopted to determine the marginal distributions necessary to the completion of the model. Two of the questionnaire respondents accepted to contribute by attempting to convert their experience into probability distributions. A simple behavioural aggregation approach was adopted, in which the second expert was presented the assessments of first expert and asked to review them. Due to the simplicity of the method implemented for the elicitation of the marginal distributions, future attempts to implement the GCBN in different settings would certainly require the definition of new marginals.

## 4. Results

This section presents the outcomes of the elicitation process conducted with the expert panel introduced earlier. After discussing the findings of the framework developed for the elicitation of dependence (Section 4.1), the marginal distributions are defined and incorporated into the model (Section 4.2).

### 4.1. Dependence Structure

#### 4.1.1. Individual Assessments

Five correlation matrices were obtained based on each expert’s responses to the ‘main’ section of the questionnaire. As illustrated in Figure 7 (for numerical values, see Appendix C), experts A, D and E indicated the prevalence of specific relationships within the network. For instance, expert D suggested the existence of one or two main predictors of each component’s condition, such as ‘Age’/‘Fans’ ($r_{1,7}=0.882$). However, the evaluation of high correlations raised problems during the elicitation and these experts were asked to review their responses multiple times to make them valid (cf. Section 2.2). As a matter of fact, and despite understanding the mathematical concepts underlying the ‘validity’ of his answers, expert A claimed that the bounds limited his ability to reflect his experience numerically. Moreover, as observed in the next section, the lack of nuance in some of the experts’ assessments strongly penalized them in the d-calibration.

In the next section, the experts’ answers to the seed questions and their respective d-calibration scores are introduced.

#### 4.1.2. Dependence Calibration

To obtain a unique set of rank correlations suitable for implementation in the Bayesian network, the individual correlation matrices presented earlier were aggregated. In this research, the d-calibration method was employed, which involved the definition of a weighted average of each expert’s responses based on their performance on a predefined set of seed variables. As illustrated in Figure 8, the results elicited from all five experts reinforce the previous observations regarding the inclination of experts D and E to assess high correlations. As outlined previously, expert D’s good understanding of probabilistic reasoning ($\rho_{1,2}≃\rho_{2,3}\gg \rho_{1,3}$) was penalized by his excessively large estimates. In contrast, experts B and C demonstrated their ability to provide moderate judgments, an important feature given the sensitivity of the rank correlations for values of $P_{c}$ around 0.5 (cf. Section 2.2), hence resulting in higher calibration scores.

Table 2 contains the d-calibration scores computed from the correlation matrices. Clearly, two groups of experts arose: whereas experts B and C obtained high d-calibration scores (≥0.85), experts A, D, and E obtained lower scores (dCal<0.66). Interestingly, all experts were ‘better’ calibrated than in other studies implementing dependence calibration (e.g., [20,46]). This observation should be taken cautiously due to the relatively small number of seed variables used, as well as the fact that they are not related to the problem at hand [20,53].

Experts were asked to evaluate the degree of comfort perceived in the assessment of probabilities. To that end, the 1–5 Likert scale shown in Table 3 was used for the statement: *I felt comfortable assessing probabilities*, whose answers are shown in Table 2. In line with the d-calibration scores, experts B and C demonstrated confidence in their assessments, whereas experts D and E encountered difficulties translating their opinions into numerical values. While expert A appeared to express confidence in his estimates, he also expressed his discomfort during the session and (indirectly) regretted the use of *unconditional* probabilities. Both experts A and D perceived the questions as ‘vague’ and the use of unconditional probabilities of concordance inappropriate, since information about one variable does not allow one to draw general conclusions about the state of others.

#### 4.1.3. Decision Makers

Expanding upon the previous analysis of d-calibration scores, this subsection aims to design and assess various combinations of the experts’ judgments, or decision makers (DMs). Two distinct DMs were subjected to evaluation: the equal weights decision maker (EWDM), defined as the average of the experts’ correlation matrices, and the global weights decision maker (GWDM), determined by a weighted average of the matrices. In the GWDM, each expert’s weight corresponds to its respective (normalized) d-calibration scores. To effectively compare the performance of these decision-makers with the respondents’, their calibration scores were computed and included in Table 2. Encouragingly, both decision makers outperformed all but the highest scoring expert (C), whose score slightly surpassed that of the global weights decision maker. Notably, the comparison between the GWDM and the EWDM did not exhibit a significant difference in performance, in line with the findings in [46]. This results from the fairly high scores obtained by all experts and the absence of an outlier.

To observe whether the gap between equal and global weights decision makers widens in the presence of an outlier, a poorly calibrated expert was added to the actual experts panel. This dummy expert’s correlation matrix for the seed variables was as follows:$$\begin{array}{c} R_{outlier}=\left[\begin{array}{ccc} 1 & 0.95 & 0.95\\ 0.95 & 1 & 0.95\\ 0.95 & 0.95 & 1 \end{array}\right] \end{array}$$
which is definite positive and performs significantly worse than the lowest-scoring expert (D): $dCal_{outlier}=0.311$. The d-calibration scores of both decision makers, computed using the new pool of experts, can be found in Table 4. The addition of an outlier notably affected the performance scores of both DMs; the EWDMs’, however, decreased more than twice as much as the GWDMs’. In the former, a minor weight is attributed to the new expert while the best performing experts (B and C) still predominantly defined the correlation matrix, whereas in the latter, the dummy’s (poor) assessment highly influenced the outcome.

Next, the existence of a ‘best’ decision maker was investigated, i.e., a combination of the experts that maximizes the calibration score. Given the gap between experts B, C, and the rest, it came with no surprise that the optimized DM (optDM in Table 2 and Table 4) is merely a weighted average of the former’s correlation matrices. This new decision maker was significantly better calibrated than the GWDM, with $dCal_{GWDM}=0.897$ and $dCal_{optDM}=0.968$. Nonetheless, we recall the limitations of the aggregation approach: the seed variables are completely unrelated to the research’s topic. Therefore, the d-calibration scores hereby assess the experts’ familiarity with probability (normative expertise), but do not provide evidence on their substantive expertise [52]. Defining the optimal DM based solely on this criterion could therefore decrease the performance of the decision maker, unveiling an opportunity for future work enhancing the method.

Previous applications of dependence calibration in academia indicated that a larger weighing pool results in the definition of more consistent decision-makers [46]. To evaluate the robustness of the DMs constructed previously, we were interested in the spread in calibration scores across the different combinations of a given size, similarly to the analysis in [46]. For an expert group of five individuals, this experiment consists of computing the d-calibration scores of all the possible combinations of experts of sizes ∈{1;2;3;4;5}.

Figure 9 illustrates the results of that experiment, where the x-axis represents the size of a given combination. For instance, let us consider x=4. The possible combinations of four experts are {A, B, C, D}, {A, B, C, E}, {A, B, D, E}, {A, C, D, E}, and {B, C, D, E}. Each of those are represented by a dot, with the y-value representing the d-calibration scores obtained by the said combination. The experiment was conducted in a similar manner after adding our dummy expert (outlier) to the panel, hence increasing the panel size to 6. Figure 9 depicts a convergence of the d-calibration scores towards higher average values for a larger experts pool, a phenomenon accentuated by the presence of an outlier. In the study’s context, where the seed variables provide little information on the expert’s substantive expertise, the risk and impact of including outliers when using the optimized DM are significant. Because the GWDM does not perform significantly worse than the optimised DM, the former was used to define the dependence structure of the GCBN. The network’s final dependence structure is shown in Figure 10.

### 4.2. Marginal Distributions

To complete the development of the network, the marginal distributions associated with each variable were defined. We recall that the model contains eight variables, presented in Section 3:‘AHU Age’: continuous. Defined on $R_{+}^{*}$.‘Maintenance interval’: continuous. Defined on $R_{+}^{*}$.‘Design & Construction quality’: discrete. Takes values between 1 (very poor) and 5 (excellent).‘Filters’, ‘Fans’, ‘Coils’, ‘Plumbing supply elements’ and ‘Electrical supply elements’: discrete. Assessed on the 1–6 scale defined in NEN 2767 [9].

In accordance with Section 3, the marginal distributions, presented in Table 5, were determined by consulting individually two of the five questionnaire respondents.

All in all, the completed GCBN was built in UniNet for illustrative purposes and is displayed in Figure 11. The final model, however, was implemented in PyBANSHEE.

## 5. Discussion

In the literature, one of two approaches are often adopted to validate a Bayesian network: the model’s predictions are compared to empirical data (when available); or experts, who contributed or not to the model creation, are asked to assess the model’s output when subjected to a set of scenarios [54,55,56]. Clearly, the use of data was excluded in this research, simply because they were unavailable. Therefore, the current model was subjected to three hypothetical scenarios to assess the model output’s logic:Scenario 1: old AHU, frequent maintenance;Scenarios 2/3: excellent Design & Construction quality, recent/old AHU.

**Scenario 1: old AHU, frequent maintenance.**  The first scenario consisted in the following configuration:•‘AHU age’: 40 years,•‘Maintenance interval’: 6 months,•‘Design & Construction quality’: 3.63 (mean value),⇒$X_{1}=\left(X_{0}=40,X_{1}=0.5,X_{2}=3.63\right)$.

The dependence structure elicited from experts indicates that the age of the unit and the frequency at which it is maintained overwhelmingly affect its condition. The outcome of Scenario 1 is shown in Figure 12. Unsurprisingly, the filters’ condition has significantly improved due to its connection with maintenance. However, this outcome, while consistent with our earlier assumptions, appears to be somewhat unrealistic from a physical standpoint. Expert D illustrated the relationship between ‘Filters’ and ‘Maintenance’ with the example of Schiphol airport, the Netherlands’ main international airport, where filters are replaced three to four times a year due to air pollution. In fact, experts almost unanimously (4/5) indicated that variables describing *environmental conditions* should be included in the model because of their impact on the filters’ deterioration. Clearly then, this scenario showcases the model’s disproportionate response as the probabilities associated to states 3 and above (for ’Filters’) should not be null, as demonstrated by the example of Schiphol.

Similarly, the distribution of the variable ‘Coils’ shifted to the left, reflecting an improvement from the unconditional case. This finds explanation in the dependence structure of the BN, where the correlation between ‘Coils’ and ‘Maintenance interval’ is substantially higher than between ‘Coils’ and ‘Age’ (0.686 and 0.275, respectively). Because the main mode of deterioration of the coils is by corrosion, accelerated by frost and the accumulation of particles, consistently cleaning them allows one to temper the phenomenon. Moreover, the shift in the distribution of ‘Filters’ also influences the one of ‘Coils’ since these variables are positively correlated. Interestingly, the probabilities of states 3, 4 and 5 are relatively low (0.15, 0 and 0, respectively) given the advanced age of the unit and the theoretical lifespan of the coils (∼20–25 years). For the same reason, the probability that the coils are in condition 1 (0.26) is abnormally high, indicating that the model’s capacity to handle extreme cases is limited.

Finally, the fans’ condition was inversely impacted by the input values: the probability that the component is in condition 4 has dramatically increased (from 0.39 to 0.65), again in accordance with the correlation of ‘Fans’ with ‘Age’ being higher than with ‘Maintenance interval’ (0.52 and 0.219, respectively). Failure in the fans mainly involves mechanical malfunctions such as exhaustion of the motor or failure of the bearings, whose maintenance has limited impact on their lifespan. The conditional probability that the component is in reasonable condition or better seems high (0.31) but most of this density is in state 3, which is conform with the previous comments.

**Scenarios 2/3: old AHU, very poor/excellent Design & Construction quality.**  The second and third scenarios consisted in the following configurations:•‘AHU age’: 40 years,•‘Maintenance interval’: 1.20 (mean value),•‘Design & Construction quality’: 1 (very poor, Scen. 2)/5 (excellent, Scen. 3),⇒$X_{2}=\left(X_{0}=10,X_{1}=1.2,X_{2}=5\right)$; $X_{3}=\left(X_{0}=40,X_{1}=1.2,X_{2}=5\right)$.

Scenarios 2 and 3 aim to determine whether an investment in an excellent quality installation significantly affects the long-term condition of the air handling unit, and whether that is reflected by the model’s outputs. The latter are displayed in Figure 13.

For old systems, an increase in ‘D&C quality’ evidently results in a slower deterioration for both ‘Coils’ and ‘Fans’, with a substantial share of the distributions being below the threshold values: $P(Coils\leq 3|X_{3})=0.999$ and $P(Fans\leq 3|X_{3})=0.903$, against $P(Coils\leq 3|X_{2})=0.164$ and $P(Fans\leq 3|X_{2})=0.0$ in Scenario 2. In alignment with the observations for Scenario 5, the impact of an excellent quality on the components is too high. While it is logical to witness an improvement from Scenario 2, the unit’s age (40 years) *must* translate in medium-to-high likelihoods for states 4 and 5. Conversely, the probabilities of states 1 and 2 are too high as both components (almost) have reached their theoretical lifespan. Clearly then, Scenario 2 indicates that the rank correlations associated to the edges ‘D&C quality’ → ‘Coils’ (−0.324) and ‘D&C quality’ → ‘Fans’ (−0.443) are possibly too high (in absolute values), which may partly stem from expert C’s strong assessment for D&C quality’ → ‘Fans’ (−0.795). His assessment, which is substantially higher than the rest of the experts’, strongly contributes to the final decision maker because of his excellent dependence calibration score.

All in all, the influence of the basic quality of the air handling unit’s components is correctly translated, even though some adjustments to the model’s parameters are still needed to obtain more realistic outputs. We notably observe that high-quality materials, design and construction can significantly extend the components’ lifespan.

As mentioned in Section 2, the addition of new variables is facilitated by the modular nature of GCBNs. Because Scenario 1 underpinned the necessity to include the node ’Environmental conditions’ as an input, and to illustrate the effect of the addition of that variable, let us consider a hypothetical network, which includes the variable ’Environmental conditions’ defined on the following scale (Table 6):

For illustration purposes, we assigned equal probabilities to each state (i.e., ‘Environmental conditions’ $∼U_{\left[\;[1,5]\;\right]}$). Then, consultations with the experts indicated that this factor mainly influences the deterioration of the filters, hence the creation of the edge ‘Environmental conditions’ → ‘Filters’. Although this relationship is weaker than that between ‘Maintenance interval’ and ‘Filters’, the conditional correlation associated to the new edge will be high since for two units maintained at the same frequency, the environmental conditions are a strong predictor for the conditions of the filters. Resultingly, we considered r(Env,Filt|Main)=−0.8. The resulting model, used in the next scenario, is illustrated in Figure 14.

**Scenario 4: old AHU, frequent maintenance and** ***very unfavorable environmental conditions.***  The fourth scenario consisted in the following configuration:•‘AHU age’: 40 years,•‘Maintenance interval’: 6 months,•‘Design & Construction quality’: 3.63 (mean value),•‘Environmental conditions’: 1 (very unfavorable),⇒$X_{4}=\left(X_{0}=40,X_{1}=0.5,X_{2}=3.63,X_{8}=1\right)$.

Figure 15 illustrates the conditional distributions of ‘Filters’ obtained in Scenarios 1 and 4. First, there is an evident change in the distribution. The discussion on Scenario 1 underlined that the probabilities of states 3, 4, and 5 could not realistically be null without information on the environmental conditions. Here, evidence of very unfavorable climatic conditions clearly resulted in a concentration of the distribution around states 3 and 4, with probabilities of 0.646 and 0.268, respectively, aligning with the example of Schiphol airport presented previously. This brief discussion demonstrates that the addition of ‘Environmental conditions’, although not rigorous, was a fairly straightforward endeavor that yielded encouraging results. Still, the previous paragraphs underlined that the developed model is not ready for practical applications.

## 6. Conclusions

To the authors knowledge, this article presents to date the first application of probabilities of concordance for the assessment of dependence in expert-based GCBNs. While the experts’ feedback indicates that this method is relevant and accessible, two elements may have influenced the validity of the elicited values. First, the closed-form equations used to compute the rank correlations from the concordance probabilities require the normal copula assumption, which often fails to reflect the behaviour of real-life systems. Second, some experts expressed difficulty assessing *unconditional* concordance probabilities, i.e., accounting for the uncertainty given that evidence on only one (parent) variable is available. Instead, the use of *conditional* concordance probabilities may help reduce the ‘vagueness’ perceived by some of the respondents.

Let Z be a vector of covariates; then, for each $z\in R^{p}$, the concordance probability between two random variables X and Y given Z=z is:$$P_{c}\left(X,Y|Z=z\right)=P\left(x_{1}\leq x_{2}|y_{1}\leq y_{2},Z=z\right).$$
with ($x_{1},y_{1}$) and ($x_{2},y_{2}$) two random draws of variables X and Y. To illustrate the practical impact of this modification, let us consider the edge between ‘Maintenance interval’ and ‘Coils’ (Figure 11). To assess P(Maint,Coils|Age), an expert would be presented the following question:

“Two buildings A and B are randomly selected among all non-residential buildings in the Netherlands. Given that the AHUs in **buildings A and B are both z years old**, and that the AHU in building A is maintained more regularly than in building B ($y_{A}\leq y_{B}$), what is the probability that the coils are in better condition in building A than building B ($x_{A}\leq x_{B}$)?”

However, for conditional concordance probabilities to be relevant, additional research should investigate the extent to which their use facilitates the elicitation and whether the protocol used to retrieve rank correlations from unconditional concordance probabilities still applies. The latter is crucial as the validity of the closed-form formulas used to retrieve rank correlations (Section 2.2.1) is not trivial in the conditional case, and is demonstrated in Appendix D. Moreover, the dependence in z can be eliminated by assuming that $P_{c}\left(X,Y|Z=z\right)$ is constant in z, similarly to assumptions formulated for conditional exceedance probabilities. All in all, the use of conditional probabilities of concordance could enhance the interpretability of the questionnaire presented to the experts, and therefore the quality of the collected assessments.

Furthermore, the application of dependence calibration in this research highlighted the challenge of selecting appropriate seed variables when few to no empirical data are available. Past studies relied on the wide availability of data in their field (e.g., [20], with traffic data) or knowledge of the ‘true’ dependence structure (e.g., [46]). However, due to the emerging nature of the method, there is no guideline for its application in data-sparse environments, sometimes constraining assessors to use equal weights decision-makers [57] or unrelated seed variables as was the case in this study. Therefore, future research should focus on assessing the potential loss of accuracy between a BN quantified with field data and another with ‘common knowledge’ information, such as precipitation or physiological data.

## Figures and Tables

**Figure 1 entropy-26-00360-f001:**
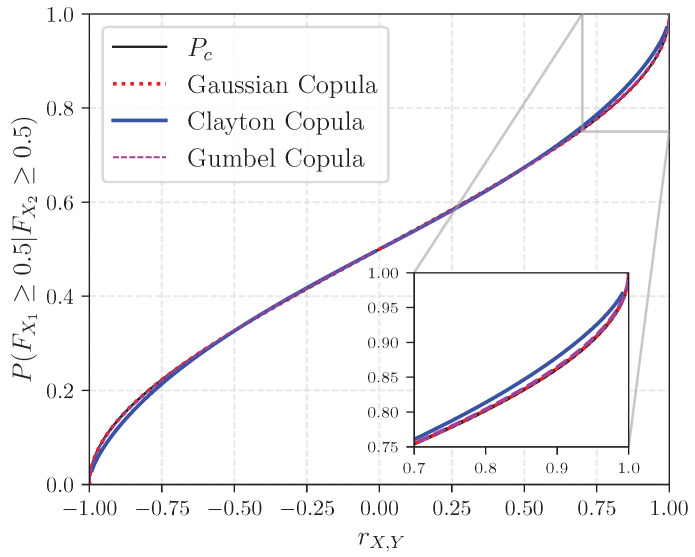
Probability of concordance and conditional exceedance probabilities as functions of the rank correlation.

**Figure 2 entropy-26-00360-f002:**
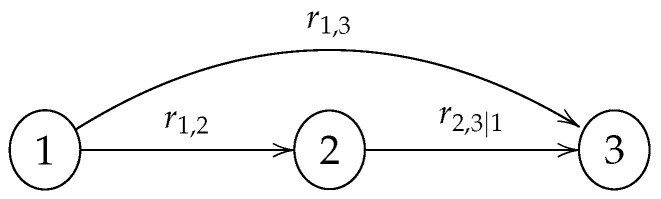
Simple Gaussian copula-based Bayesian network (GCBN) with 3 nodes.

**Figure 3 entropy-26-00360-f003:**
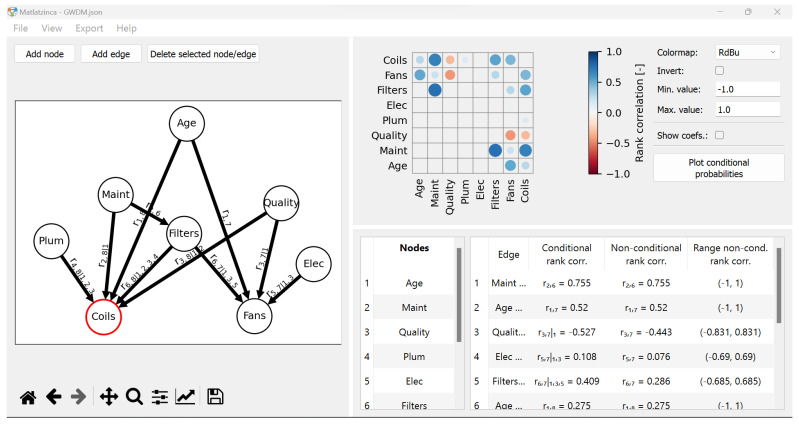
Matlatzinca graphical user interface (GUI) for the elicitation of GCBNs from experts. On the left the drawing panel, on the top-right the correlation matrix panel, and on the bottom-right the input panel.

**Figure 4 entropy-26-00360-f004:**
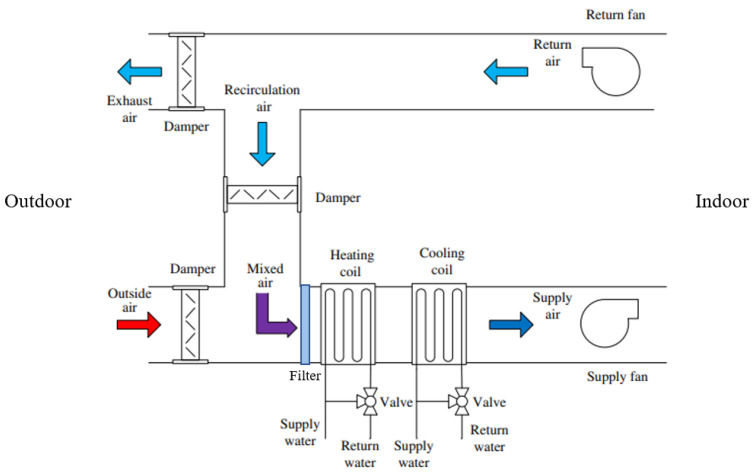
Air handling unit (AHU) with air recirculation (adapted from [47]).

**Figure 5 entropy-26-00360-f005:**
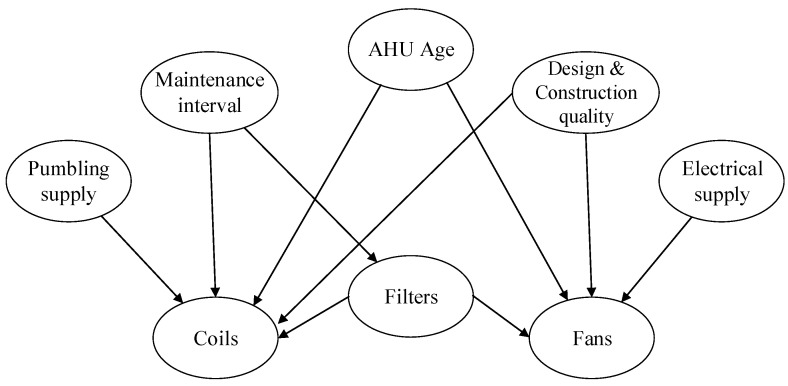
Graph structure for AHUs.

**Figure 6 entropy-26-00360-f006:**
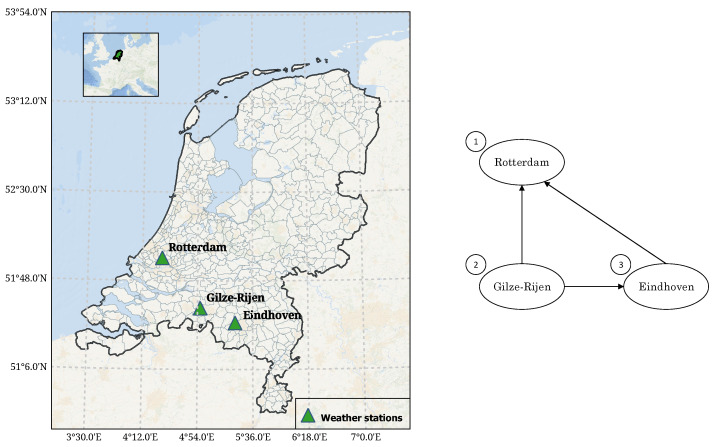
Location of the weather stations (**left**) and associated graph (**right**) used for the assessment of seed probabilities.

**Figure 7 entropy-26-00360-f007:**
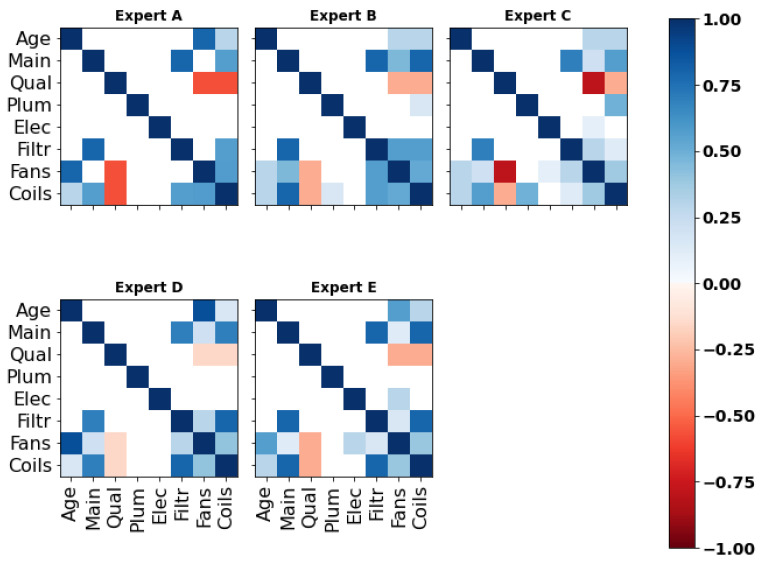
Correlation matrices retrieved from the ‘main section’ of the questionnaire.

**Figure 8 entropy-26-00360-f008:**
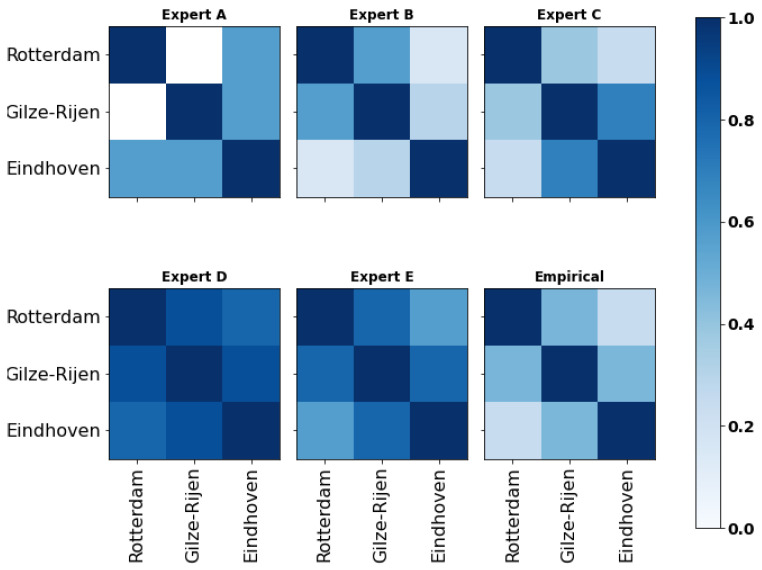
Correlation matrices retrieved from the seed questions and empirical correlation matrix.

**Figure 9 entropy-26-00360-f009:**
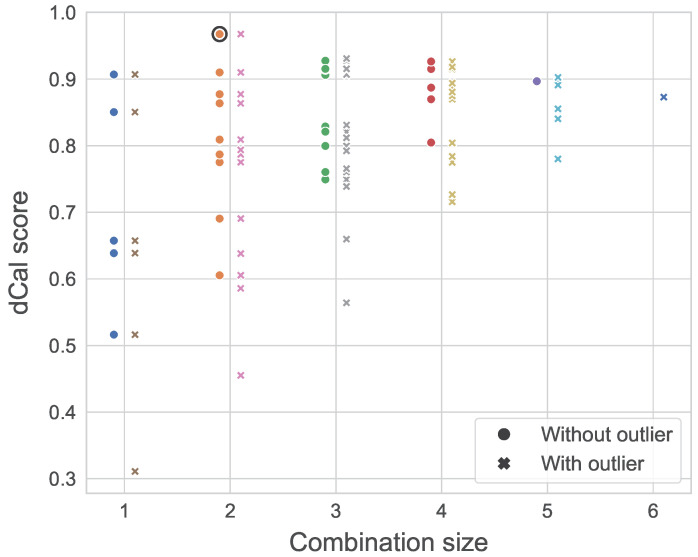
Dependence calibration scores of the global weights decision maker (GWDM) for all combinations of experts, with and without outlier.

**Figure 10 entropy-26-00360-f010:**
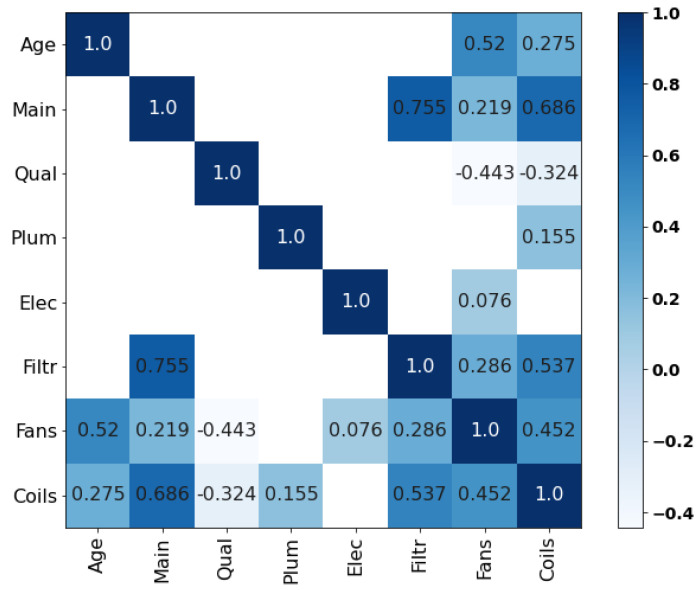
Correlation matrix implemented in the GCBN.

**Figure 11 entropy-26-00360-f011:**
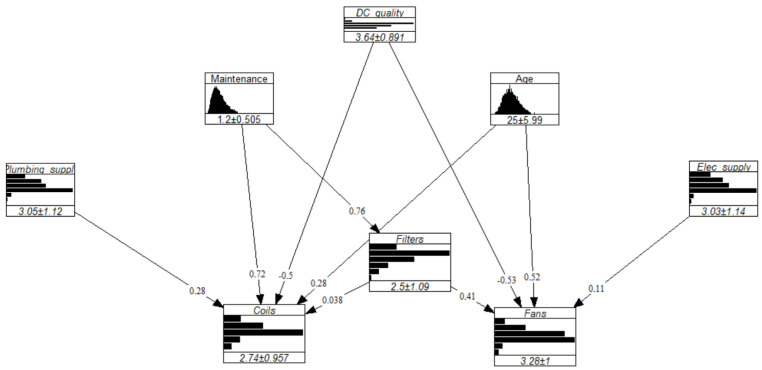
Visualization of the quantified GCBN in UniNet.

**Figure 12 entropy-26-00360-f012:**
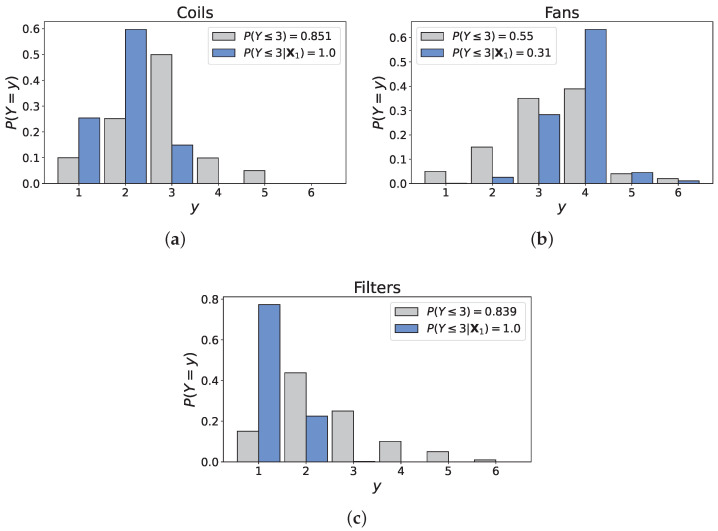
Unconditional and conditional distributions of the output variables. (**a**) Distribution of ‘Coils’. (**b**) Distribution of ‘Fans’. (**c**) Distribution of ‘Filters’.

**Figure 13 entropy-26-00360-f013:**
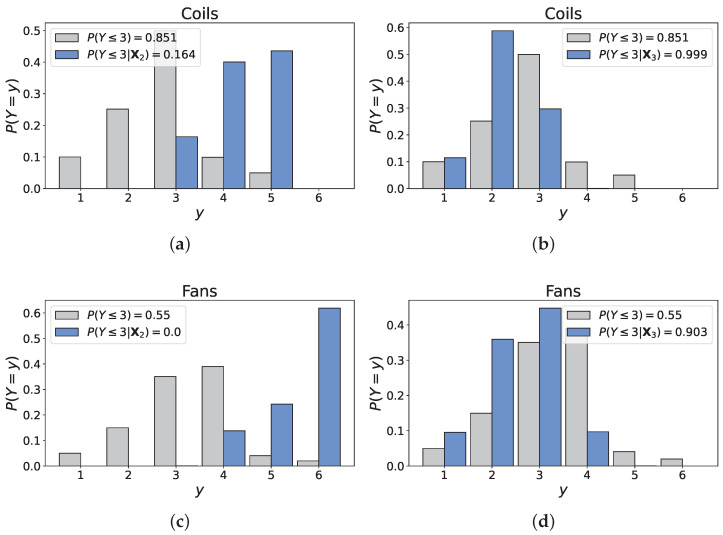
Unconditional and conditional distributions of the output variables. (**a**) Distribution of ‘Coils’ under Scenario 2. (**b**) Distribution of ‘Coils’ under Scenario 3. (**c**) Distribution of ‘Fans’ under Scenario 2. (**d**) Distribution of ‘Fans’ under Scenario 3.

**Figure 14 entropy-26-00360-f014:**
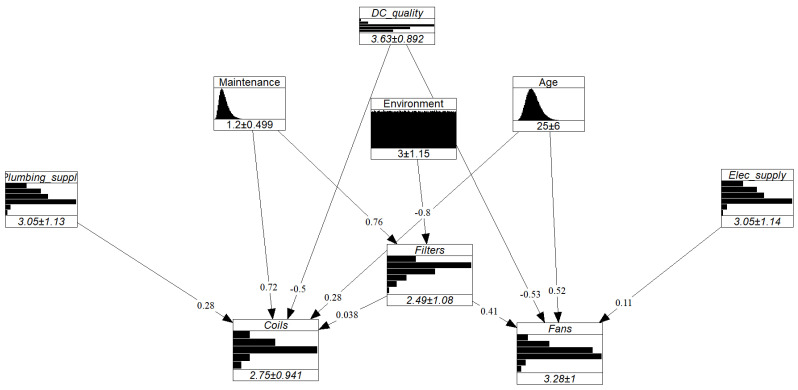
Hypothetical GCBN including the variable ‘Environmental conditions’.

**Figure 15 entropy-26-00360-f015:**
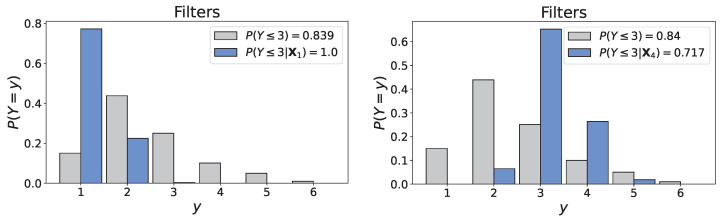
Unconditional and conditional distributions of ‘Filters’ under Scenario 1 (**left**) and Scenario 4 (**right**).

**Table 1 entropy-26-00360-t001:** Design & Construction quality of the installation scale.

Very Poor	Poor	Medium	Good	Excellent
1	2	3	4	5

**Table 2 entropy-26-00360-t002:** Experts’ dependence calibration scores and perceived degree of comfort during the elicitation.

Decision Maker	D-Calibration	Perceived Comfort
Expert A	0.639	4
Expert B	0.907	4
Expert C	0.85	4
Expert D	0.516	2
Expert E	0.657	2
EWDM	0.869	-
GWDM	0.897	-
optDM	0.968	-

**Table 3 entropy-26-00360-t003:** Likert scale used to measure questionnaire respondents’ perceived comfort.

Strongly	Disagree	Neither Agree or Disagree	Agree	Strongly Agree
1	2	3	4	5

**Table 4 entropy-26-00360-t004:** Equal and global weights decision makers’ scores with and without outlier.

Decision Maker	Without Outlier	With Outlier
Expert D	0.516	-
Outlier	-	0.311
EW DM	0.869	0.818
GW DM	0.897	0.873
optDM	0.968	-

**Table 5 entropy-26-00360-t005:** Marginal distributions of the GCBN’s variables. * std: standard deviation.

Variable	Distribution	(Mean, std *)
Age	LN(μ=3.191,σ=0.237)	(24.98, 6.00)
Maintenance interval	LN(μ=0.102,σ=0.40)	(1.20, 0.50)
D&C quality	$P{\left(X=i\right)}_{i\in [1,5]}=$ [0.01, 0.05, 0.44, 0.3, 0.2]	(3.63, 0.89)
Filters	$P{\left(X=i\right)}_{i\in [1,6]}=$ [0.15, 0.44, 0.25, 0.1, 0.05, 0.01]	(2.49, 1.08)
Fans	$P{\left(X=i\right)}_{i\in [1,6]}=$ [0.05, 0.15, 0.35, 0.39, 0.04, 0.02]	(3.28, 1.00)
Coils	$P{\left(X=i\right)}_{i\in [1,6]}=$ [0.1, 0.25, 0.5, 0.1, 0.05, 0]	(2.75, 0.94)
Plumbing supply elts	$P{\left(X=i\right)}_{i\in [1,6]}=$ [0.12, 0.2, 0.24, 0.4, 0.03, 0.01]	(3.05, 1.14)
Electrical supply elts	$P{\left(X=i\right)}_{i\in [1,6]}=$ [0.12, 0.2, 0.24, 0.4, 0.03, 0.01]	(3.05, 1.14)

**Table 6 entropy-26-00360-t006:** ’Environmental conditions’ scale.

Very Unfavorable	Unfavorable	Medium	Favorable	Very Favorable
1	2	3	4	5

## Data Availability

Data are contained within the article.

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
