# Peer review of "Elicitation of Rank Correlations with Probabilities of Concordance: Method and Application to Building Management"

_entropy, 2024, doi:10.3390/e26050360_

Round 1
Reviewer 1 Report
Comments and Suggestions for Authors
1.
Minor corrections are annotated in the attached file.
2.
I suggest writing τ instead of "tau" throughout the paper.
I suggest writing "section ..." when a section, subsection, or subsubsection is cited.
For example, section 2, section 2.1, section 2.1.1.
3.
At line 197, when Matlatzinca is mentioned for the first time, a bibliographic reference or a web-site should be cited.
4.
At lines 314-326, the names of the corresponding BN nodes should be mentioned while describing the dependencies in the case study (in the model).
5.
Sec. 5.2 is entitled "Elicitation method", but according to the contents, this section seems to be the conclusion of the paper.
I suggest renaming this section as "6. Conclusions".
6.
Appendix A and appendix B contain no text.
I suggest adding some lines explaining the contents of fig. A.1 and Table B.1, respectively in Appendix A and B.
7.
In the text of Appendix C the figures from C.1 to C.5 should be cited, with a brief description of their contents.
8.
In Appendix C the figures from C.1 to C.5 contain two images (a, b).
In the caption of every figure the content of images a and b should be explained, as done in fig. A.1.
9.
Equations A1, A2, A3 are in appendix D, so they should be D1, D2, D3.
The following equations are not numbered.
Equations A2 and A3 (D2 and D3) correspond to equations 7 and 8 in the main text.
This should be clarified by adding a sentence about that.
10.
It seems to me that table 5 is not cited in the main text.
11.
In case of resubmission of the paper, please highlight the changes by means of a different colour.

Minor corrections are annotated in the attached file.
Reviewer 2 Report
Comments and Suggestions for Authors
This paper is fine as far as it goes. Background on DAGs should be expanded; I could guess at definitions of parents and children, and the notation pa(.), but little space needs to be used to make this explicit. Also, notions like the rank correlation are almost always defined in terms of sample quantities. The paper uses them as population quantities, which is OK, but is not what readers who know about these areas expect when reading the MS.
On page 15, I don't understand the use of the word litigious; I don't think that the authors intend what this word means.
